# Catalyst-Enhancing Hydrothermal Carbonization of Biomass for Hydrochar and Liquid Fuel Production—A Review

**DOI:** 10.3390/ma17112579

**Published:** 2024-05-27

**Authors:** Waheed A. Rasaq, Charles Odilichukwu R. Okpala, Chinenye Adaobi Igwegbe, Andrzej Białowiec

**Affiliations:** 1Department of Applied Bioeconomy, Wrocław University of Environmental and Life Sciences, 37a Chełmońskiego Str., 51-630 Wrocław, Poland; waheed.rasaq@upwr.edu.pl (W.A.R.); chinenye.igwegbe@upwr.edu.pl (C.A.I.); 2UGA Cooperative Extension, College of Agricultural and Environmental Sciences, University of Georgia, Athens, GA 30602, USA; charlesokpala@gmail.com; 3Department of Chemical Engineering, Nnamdi Azikiwe University, P.M.B. 5025, Awka 420218, Nigeria

**Keywords:** thermal conversion process, feedstock, hydrochar and bio-oil, catalyst, hydrothermal carbonization

## Abstract

The research impact of catalysts on the hydrothermal carbonization (HTC) process remains an ongoing debate, especially regarding the quest to enhance biomass conversion into fuels and chemicals, which requires diverse catalysts to optimize bio-oil utilization. Comprehensive insights and standardized analytical methodologies are crucial for understanding HTC’s potential benefits in terms of biomass conversion stages. This review seeks to understand how catalysts enhance the HTC of biomass for liquid fuel and hydrochar production, drawing from the following key sections: (a) catalyst types applied in HTC processes; (b) biochar functionality as a potential catalyst; (c) catalysts increasing the success of HTC process; and (d) catalyst’s effect on the morphological and textural character of hydrochar. The performance of activated carbon would greatly increase via catalyst action, which would progress the degree of carbonization and surface modification, alongside key heteroatoms. As catalytic HTC technology advances, producing carbon materials for thermochemical activities will become more cost-effective, considering the ever-growing demands for high-performance thermochemical technologies.

## 1. Introduction

Industrial output and economic growth are propelled by energy. The global quantity of energy consumption rises annually with population and intense urbanization. To meet global needs, especially in developing nations, global energy consumption is projected by the end of the century to exceed 84,000 metric tons [1,2]. In daily production processes, organic solid waste (OSW) as a potential resource loses its original value [2]. Since conventional fossil fuels still comprise a significant amount of the market today, finding substitutes for fossil fuels is essential. Thus, OSWs, as a fuel with a bright future, remain a significant energy conservation candidate [3]. Considered a waste product of human existence/production, OSWs are primarily separated into two groups as follows: (a) non-lignocellulosic wastes (such as sludge, manure, and digested food waste), and (b) lignocellulosic wastes (such as yard waste and agricultural waste) [4,5]. In urban areas, digested food waste would potentially be an OSW source of bio-renewable energy production [6,7], able to lessen environmental issues and fossil fuel reliance.

In general, biochemical and thermochemical processes can convert organic waste into energy. Anaerobic digestion is the most used biochemical approach for generating biogas for power plant purposes. High levels of volatile fatty acids and ammonia, however, can easily cause anaerobic digestion to become unstable and inefficient [8], and the release of biogas digestate with large volumes and high nutrient contents also poses a significant environmental risk. More crucially, because it can represent an unanticipated risk to humans through food chains, the proliferation of new pollutants, in particular antibiotics and antibiotic resistance genes, has sparked worry across the globe [9]. In contrast, thermochemical methods and biochemical methods, (gasification, pyrolysis, hydrothermal carbonization, and combustion) can efficiently convert biomass and manure into energy in a short time [4,10,11,12]. Previously, the most common thermal processes, like pyrolysis and incineration, appeared less effective in terms of energy recovery and value-added products, given the high moisture content of emergent organic waste. The presence of dangerous bacteria in waste presented certain challenges for technologies, including limited processing efficiency and the creation of secondary waste pollutants [13,14,15]. To successfully convert OSW into energy, one of the main priorities is to find a suitable method for OSW processing. Hydrothermal carbonization (HTC) technology is discovered to be an effective method for OSW treatment since HTC could reduce the need to dry the feedstock, providing a carbonized solid coupled with an aqueous phase rich in nutrition [13,14,15,16]. Biofuels and valuable chemicals can be produced from renewable resources like waste materials, woody and herbaceous biomass, forestry residues, crops, and algal biomass through HTC treatment [17]. A series of reactions known as decarboxylation, dehydration, condensation, and aromatization take place during HTC. According to earlier research, produced hydrochar can be used for energy storage, environmental remediation through the adsorption of heavy metals, CO_2_ capture in the agriculture sector, and the production of alternative fuel feedstock for the steel and cement industries [18].

The research impact of catalysts on the HTC process remains an ongoing debate, especially the quest to enhance biomass conversion into fuels and chemicals, which requires diverse catalysts to optimize bio-oil utilization [19,20]. Different types of homogenous [21,22,23] and heterogeneous catalysts [24] employed in HTC would enhance the product yield and associated properties [25]. Furthermore, catalysts reducing tar and char formation would help progress the efficiency of the liquefaction process [26]. The water–gas shift reaction can be sped up through the use of catalysts, which can also increase the liquid yield [27,28]. Alkali catalysts also lessen the production of char and tar. Song and colleagues [29] reported that the yield of crude considerably increased from 33.4% to 47.2% when 1.0 wt% Na_2_CO_3_ was added to wood biomass, and other workers reported an increase in the oil yield from 17.88 wt% without a catalyst to 34.85 wt% with the addition of K_2_CO_3_ at 300 °C [30]. At 280 °C and concentrations of 0.235–0.94 M K_2_CO_3_, it was demonstrated that char formation was decreased while the liquid yield increased from 17.8% to 33.7% [31]. High-efficiency and inexpensive catalysts that undergo thermal conversion treatment must be viewed as an economically viable tactic that can successfully compete in the current energy market. Understanding, therefore, the foundation of catalysts, like magnesium oxide (MgO), is crucial. Calcium-based materials are considered another attractive and emerging aspect. These materials are thought by some to be reasonably priced catalysts that can provide catalytic HTC. Commercial lignocellulosic biomass could also be used in fast catalytic pyrolysis, which would involve a particular scale like a circulating fluid-bed reactor facility. Furthermore, the foundation sites of MgO would promote ketonization and aldol condensation reactions, resulting in an adequate production of hydrogen bio-oil [32]. For instance, Oliveira and colleagues reported that a bimetallic Pt/Rh catalyst, supported on carbon black and water, obtained at the lowest carbonization temperature, was eventually used to achieve a maximum H_2_ yield of 98.7 mmol H_2_ gTOC^−1^ [33]. Rather than 250 °C to 300 °C, the temperature for the employed catalyst ranged from 150 °C to 180 °C. In this context, and given the thermodynamic constraints, lower process temperatures would favor a higher conversion of syngas into liquid fuels [34]. Furthermore, biochar-based catalysts, which are categorized as non-graphitizable, have drawn a lot of interest recently for their ability to produce biodiesel from microalgal lipids [35]. Moreso, the induction of chemical activation also played a major role in the context of thermal conversion products.

After impregnating biochar or hydrochar with one or more chemical agents (oxidizing agents, alkaline solutions, acids, etc.), an activation process in a fixed-bed reactor with a nitrogen flow rate is carried out [36]. However, phosphoric acid (H_3_PO_4_), sodium hydroxide (NaOH), zinc chloride (ZnCl_2_), potassium hydroxide (KOH), and potassium carbonate (K_2_CO_3_) are the most-used chemical activating agents for the chemical activation process [37,38,39]. At low temperatures, ZnCl_2_ can penetrate the inner of the biomass and can remain liquid during the entire thermal process below 700 °C. Consequently, ZnCl_2_ is dispersed uniformly throughout the biochar’s matrix. ZnCl_2_ could be eliminated to produce a well-developed microporous biochar. Additionally, given that ZnCl_2_ has a strong capacity to dehydrate at high temperatures, it may lower the carbonization temperature of biomass components. Moreover, ZnCl_2_ inhibits the formation of tar and modifies the pathways by which biomass decomposes [40]. Tevfi and colleagues recently used a high-pressure (40 MPa) autoclave reactor at three different temperatures (255, 275, and 355 °C) to liquefy Syrian mesquite stem in order to produce bio-oil in supercritical acetone and methanol with and without (zinc chloride, sodium hydroxide) a catalyst. They discovered that the peaked conversion of 77.96% and the liquid yield of 49.67% were attained in acetone at 295 °C with zinc chloride present [41]. Furthermore, biological catalyst systems which include free lipase, traditional immobilized lipase, and lipase immobilized on magnetic nanoparticles are of growing interest in the research. Although free enzyme catalysts in biological catalyst systems offer many advantages over chemical catalysts, the high cost of the enzymes and their non-reusability contribute to the high cost of biodiesel production. Considering that immobilized enzymes can be recycled, more favorable attention is required because they aggregate well, and, like silica-coated magnetic nanoparticles, are the ideal carrier for immobilizing enzymes [42]. Overall, the current kinetics reaction and statistical methods are significantly affected by the experimental data used for the calibration. Indeed, there could be proposed relationships between the HTC operating conditions and the properties of emergent phases.

A summary of reviews involving hydrothermal treatment and biomass conversion, along with catalysts/catalytic processes within the recent decade is shown in Table 1. Most reviews appear to focus on the catalyst as an enhancement tool of HTC products like biochar (solid from other thermal conversion process), hydrochar (solid from HTC), and bio-oil [37,43,44,45], including how catalyst types and their mechanisms determine the bio-oil yield [19,46,47,48]. There are some reviews that have shown activated biochar as a catalyst, and how the types favor the process that leads to an emergent end product [49,50,51]. While many reviews focus on the applications, the physical and chemical properties of the products, and the chemistry of the process, they also present the general knowledge of HTC [52,53,54,55]. Given the rapid development of experimental studies into the influence of catalysts in HTC, continued efforts using literature synthesis are necessary to highlight the strengths (of catalysts) and the generation of fuels, including value-added chemicals from different feedstock. In addition to enhancing the potential benefits of the HTC of biomass conversion stages, the corresponding standard analytical methodologies are crucial for understanding the HTC’s potential benefits in terms of biomass conversion stages. To supplement existing information, therefore, this current review seeks to understand how catalysts enhance the HTC of biomass for liquid fuel and hydrochar production, drawing from the following key sections: (a) catalyst types applied in HTC processes; (b) biochar functionality as a potential catalyst; (c) catalyst increasing the success of the HTC process; and (d) the catalyst’s effect on the morphological and textural character of hydrochar. To provide a pictorial viewpoint of this work, key stages in the application of catalysts in the thermal conversion process, from feedstock selection, catalyst types, treatment methods, and analytical methods of output, are presented in Figure 1.

## 2. Catalyst Types Applied in the HTC Process

Catalysts play a crucial role in either the thermal conversion process or ex situ upgrading via impacting the overall multiscale design of these processes [56,57]. To ensure the best catalyst combination and process design, the catalytic upgrading of HTC necessitates a thorough understanding of the chemical reactions that result in the desired products, as well as the identification of the catalyst species that would favor these transformations at various process configurations [57,58]. This is a complex challenge due to the three following reasons: (A) There are many components present in the product, derived from the sequential reactions happening during the biomass thermal conversion process. For instance, a typical bio-oil contains more than 300 oxygenated compounds [58]. Selecting a catalyst that can convert these species selectively into the desired products with minimal by-product formation is challenging. (B) The feedstock composition is frequently liable to large variations because of non-homogeneities in the material feed and process conditions. It is difficult to find a catalyst that can tolerate these alterations, but it is crucial for the various thermal and upgrading processes [56]. (C) Finally, the most advantageous combination of operating conditions and catalyst type is a significant difficulty in the development of multi-scale processes, largely due to the catalyst’s “apparent” performance being reliant on the kind of operating parameters and reactor. The first part of this section provides a thorough overview of the state of our understanding regarding various upgrading catalysts and the kinetic pathways they are associated with. Different catalyst impacts and associated operating conditions are then reviewed [56]. Generally, a variety of fuels including methane, hydrogen, ethanol, and chemicals such as fructose, sorbitol, glucose, lactic, and levulinic acid can be obtained from the catalytic conversion of organic material [58].

Reactions involving catalytic transesterification can employ biological or chemical catalysts. Two catalytic routes are recognized for improving the properties and yield of the HTC products, including (a) the application of homogenous catalysts, such as alkali or organic acid catalysts, and (b) the application of heterogeneous catalysts, such as supported metals, molecular sieves, altered molecular sieves, insoluble inorganic salts, transition metal oxide, and others. The homogeneous catalysts are comprised of acid and alkali catalysts. Solid acid, base, biomass waste-based, acid-base bifunctional, and nanocatalysts are all included in the heterogeneous catalyst [59].

### 2.1. Homogeneous Catalysts

Homogeneous catalysts with a broad range of applications that have been investigated for the HTC include (1) alkaline compounds like carbonates and hydroxides with K, Na, and Ca forms; (2) organic acids like acetic and formic acid; and (3) inorganic acids like sulfuric acid. The homogenous catalysts used in the HTC are soluble in water at room temperature. In certain situations, homogeneous catalysts can process liquids without experiencing coking, making them cost-effective [60].

Nevertheless, homogenous catalysts have certain disadvantages as well. When employing homogeneous catalysts [61], the catalyst recovery process requires energy-intensive and expensive separation stages. Another drawback is that the homogeneous catalysts are corrosive, which is an important factor to consider when choosing the materials for the HTC reactor design [62]. Since the catalyst selection can reduce production costs, it plays a significant role in the synthesis of HTC products. The amount of free fatty acids (FFAs) in the feedstock oils determines the type of catalyst that should be used. The use of homogeneous catalysts is the first conventional technique for producing biodiesel. Homogeneous catalysts exist in the same phase as their reactants. On the other hand, homogeneous catalysts can be classified into two categories: homogeneous acid catalysts and homogeneous alkali catalysts. Since the reaction is fast and the reaction conditions are moderate, homogeneous alkali catalysts such as CH_3_ONa, CH_3_KO, KOH, and NaOH are the most widely used industrial catalysts in the industrial transesterification process for the production of biodiesel [63]. When extra-pure virgin oils are used, with FFA contents and acid values, respectively, of less than 0.5% and 1 mg KOH/g, homogeneous alkali catalysts should offer superior purity/yield, which is why enhancing the quality and, at the same time, maximizing the bio-oil yield has been crucial in catalysts’ performance. In HTC, a variety of heterogeneous [21,22,64,65,66,67] and homogeneous [58,68] catalysts have been employed to enhance the yield and characteristics of bio-oil. Previously, a homogenous catalyst like Na_2_CO_3_ seems to dominate in the majority of HTC studies of lignocellulosic biomass to increase the bio-oil yield. But, while some studies found the yield to decrease [69], in the report of Long and colleagues, the HTC of bagasse in subcritical water MgMnO_2_ was assessed. The intensification effect of the MgMnO_2_ was investigated, where the product distribution and composition of volatiles and residue were compared. The result demonstrated the relative content of furfural of 2-methyl-2-cyclopenten-1-one, (250 °C in 5.02 min), 2-hyd roxy-3-methyl-2-cyclopenten-1-one (250 °C in 7.67 min), and the significant increase in their derivatives. These compounds generally come from the Aldol condensation of the bagasse carbohydrate HTC product, which can be enhanced by the alkali catalyst [67]; others reported that adding Na_2_CO_3_ to various algal strains would increase the bio-oil yield [23,70]. Shakya and colleagues studied the bio-oil yield of *Nannochloropsis* with Na_2_CO_3_, and found this to be considerably lower at 250 °C than it was at higher temperatures. This kind of algal strain’s high protein content was most likely the cause of this. Peptide bonds in proteins and glycosidic bonds in carbohydrates are more stable at lower temperatures. As a result, proteins hydrolyze slowly at low temperatures. However, proteins hydrolyze more readily at temperatures between 300 and 350 °C, which increases the amount of bio-oil produced. Because of the higher protein conversion at 350 °C, *Nannochloropsis* produced a larger yield. Additionally, this demonstrates that, at higher temperatures, the relative abundance of nitrogenous compounds increases [25]. The different ways that lipids, proteins, and carbohydrates liquefy when Na_2_CO_3_ is present could account for this discrepancy in the results [25]. To further elaborate the above discourse, Table 2 shows the catalyst (homogeneous and heterogeneous) types by feedstock, HTC operating conditions, and product yield [28,65,67,68,71,72,73,74,75,76,77,78,79,80,81,82]. The homogeneous catalysts like KOH, Na_2_CO_3_, CH_3_COOH, and others display significant variability in product yields, underscoring the influence of the catalyst type and feedstock compatibility. Therefore, understanding the catalyst choice/type, the selection of feedstock, and the corresponding parameters involved in HTC treatment helps in achieving higher quality products.

### 2.2. Heterogeneous Catalysts

Typically, heterogeneous acid catalysts exist in a solid state and function at distinct stages within the liquid reaction mixture. A wide range of solid catalysts have been used to produce biodiesel during the past ten years. The benefits of heterogeneous catalysts’ resistance to water and the amount of FFA in feedstock are making them more important for the production of biodiesel [19,83]. Heterogeneous acid catalysts can overcome the primary issues related to toxic effects and vessel corrosion when compared to homogeneous acid catalysts [19]. These catalysts allow biodiesel production from inexpensive low-quality feedstocks without acid pretreatment because they are insensitive to the high FFA and water content in the feedstock oils [19]. Though BaO is toxic and easily soluble in methanol or ethanol, CaO and BaO are typically stronger than MgO [84]. Given its superior availability, activity, selectivity, and low solubility in methanol, calcium oxide (CaO) is considered an affordable, easily accessible, and highly effective heterogeneous catalyst that requires moderate reaction conditions [83]. Furthermore, when producing industrial biodiesel, it remains extremely stable for longer periods. Das and colleagues [85] generated biodiesel with the oil of *Scenedesmus quadricauda* algae and a cobalt-doped CaO catalyst.

Metal catalysts are useful in the manufacturing of jet fuel, diesel, oil, and other fuels, but they also hold promise for the next wave of green energy technologies. For instance, liquid hydrazine (N_2_H_4_) decomposes at an ambient temperature to form N_2_, H_2_, and NH_3_ over a commercial Ir/-Al_2_O_3_ catalyst, which is already utilized as a propellant to modify the satellite orbit and attitude [86]. Solid catalysts such as silica-alumina, zeolites, and supported metals are so far preferred as catalytic materials for improving associated hydrocarbon fuels and bio-oil yields, as mentioned in the section of the list of catalyst types [87]. Feedstock algae or other biomass types can involve both homogeneous and heterogeneous catalyst types, which might differ in terms of HTC operating conditions and product yield. Table 2 also reveals that there are instances where char and gas were not analyzed after HTC [28,68,71,72,75,80,81]. Heterogeneous catalysts such as Ce/H-ZSM−5, H-ZSM−5, and others exhibit diverse bio-oil, char, and gas yields, emphasizing the sensitivity of the outcomes to catalyst composition and operating conditions. Xu and colleagues study the HTC of *Chlorella pyrenoidosa* with the addition of Ce/HZSM-5 and HZSM-5 to analyze the chemical groups and components of *C. pyrenoidosa* bio-oil. The results showed that the effects of Ce/HZSM-5 were superior to that of HZSM-5 due to its highly dispersed Ce_4_O_7_ with trivalent and tetravalent cerium in the zeolite skeleton channel, smaller particle size, larger specific surface, and significantly enhanced Lewis acid active center when compared with HZSM-5. The components of the bio-oils revealed that feedstock contains organic compounds with C4–C16 oxygen, such as aldehyde, ketone, acid, ester, and some chemicals that contain nitrogen, which originate from the protein in *C. pyrenoidosa.* Additionally, it has a higher heating value, which may be explained by the presence of more hydrocarbons such as cyclane derivatives, benzene derivatives, and alkene derivatives. Their findings demonstrate Ce/HZSM-5’s strong catalytic properties and potential applications [80]. Furthermore, other reports discuss the production of biodiesel using heterogeneous acid catalysts. However, since the heterogeneous acid catalyst is typically hydrophilic, the water that is created during the esterification of fatty acids will reduce its activity. This is because the acid catalysis of these inorganic oxide solid acids takes place in the acidic hydroxyl groups (-OH), which function as potent Brönsted acid sites. In the presence of water, the hydration of -OH would lessen the acid strength of these [88,89,90]. Additionally, the low acid site concentration, microporosity, and the hydrophilic nature of the catalyst surface raise issues in terms of heterogeneous acid catalysts. It was recently reported that a novel class of solid acid catalyst based on sulfonated carbon showed promise in producing biodiesel [88].

In summary, Table 2 underscores the importance of tailored catalyst selection based on feedstock and operating parameters to achieve optimal outcomes in HTC processes, as evidenced by the distinct performances detailed for each catalyst across different works.

## 3. Biochar Functionality as a Potential Catalyst

Recently, a range of solid waste materials (including egg shells, fly ash, and fish bones) have been used as feedstocks for the preparation of hydrochar to be used as affordable catalysts to reduce the high cost of catalyst synthesis [91]. Biochar is a low-cost/carbon-rich material produced via thermochemical degradation. When compared to other commercially available solid-based catalysts, biochar is highly recommended due to its benefits over other catalysts involving enhancing the quality and yield of the thermal conversion process of several feedstock types [92]. The utilization of biochar as a carbonaceous catalyst or support in the production of biodiesel holds great potential, owing to its inexpensive cost, the presence of surface functional groups, and its relatively high surface-to-volume ratios [93]. Biochar is used as a heterogeneous catalyst or support because it is inexpensive, can be tailored to specific functional groups, has a large surface area, and is perfect for producing biodiesel, as shown in Figure 2. Moreso, it has an environmentally friendly nature, and good thermal, chemical, and mechanical stability [94]. Numerous scientists choose to investigate biochars because they are cheap, reusable, and environmentally friendly catalysts. According to Ormsby and colleagues, during the simultaneous reactions of transesterification and esterification of non-edible oils, recyclable biochar-based catalysts demonstrated better activity when compared with conventional acid catalysts [95]. Chang and colleagues demonstrated that the addition of inorganics (K and Fe) improves the catalytic activity of biochar [40], and the adsorption of metal precursors towards the synthesis of biochar-supported metal catalysts is facilitated by the presence of functional groups on the surface of biochar [51]. Furthermore, Chi and colleagues reported a study in which the biochar was treated with 10 M KOH, which resulted in catalysts with the highest catalyst activity for biodiesel production produced from canola oil due to their increased surface area and acid density [35]. Sulfonated biochar with ethanol at 60 °C demonstrated 77–88% fatty acids conversion from waste vegetable oil [96], which can be used directly/in combination with petroleum diesel in most diesel engines [51].

The merits of biochar-based catalysts come from the production process, which makes it straightforward and profitable due to the availability of sustainable feedstock, and the physicochemical properties of biochar can be easily modified through different activation techniques. Essentially, surface functional groups, the presence of inorganic species, and the hierarchical structure derived from biomass are among the biochar features that make catalysts superior in diverse applications [35,40]. Due to those mentioned above, small-scale research on biochar as a catalyst remains scanty, and, to fully grasp its potential, particularly for long-term feasibility and economic viability, more extensive research is imperative. Furthermore, to make a significant advancement in the field of biochar-based catalysts, new technologies must be substituted for outdated ones, such as equipment, activation strategies, and conversion technology.

## 4. Catalyst Increasing the Success of the HTC Process

It is well-established that catalysts occupy crucial space in hydrothermal carbonization (HTC), providing numerous benefits, from enhancing bio-oil yield, improving biomass conversion, increasing biofuel flow properties, reducing biofuel heteroatom content, to lowering the required temperature for optimal biofuel yield [47]. Alkali, acids and metal salts, due to their cost-effectiveness, thrive in homogeneous catalysis during the hydrothermal liquefaction process. However, their recovery and corrosiveness challenges would limit their application, which might shift the focus towards heterogeneous catalysts [97] which are believed to offer high catalytic activity, low corrosion rate, and easy recovery compared to homogeneous catalysts [48]. Several homogeneous and heterogeneous catalysts have been shown to increase bio-oil yield during the hydrothermal liquefaction process, wherein heterogeneous catalysts revealed higher conversion efficiency over homogeneous catalysts [87,98]. An iron/nickel oxide nanocomposite resulted in a maximum bio-oil yield of 59.4 wt%, surpassing the 50.7 wt% achieved without the catalyst. At a temperature of 320 °C, 60 min of residence time, and 1.5 g of catalyst dosage, the maximum bio-oil yield was achieved [98]. HTC on various biomasses has enhanced the efficiency of liquefaction by catalysts, wherein alkali catalysts (such as KOH, NaOH, Na_2_CO_3_, and K_2_CO_3_) were particularly applied to wood [99,100], bark [101], EPFB (palm fruit bunch) [102], switchgrass [103], and algae [69]. Thus, alkali catalysts are crucial in enhancing biomass conversion, increasing bio-crude output, and improving bio-crude quality by elevating hydrogen content and decreasing oxygen concentration. Besides liquid yield and biomass conversion, where catalytic activity follows the order of K_2_CO_3_ > KOH > NaOH in some studies [100,102], Zhao and colleagues showed that the pore and surface area volume of the biochar increased significantly following chemical activation treatment with KOH [91]. Additionally, potassium carbonate has been shown to serve as the catalyst where hydroxides induce more severe equipment corrosion [58]. HTC product distribution, including bio-oil, gas, and char, has also received great interest. The total oil yields of cotton stalk, wheat straw, and corn stalk were less than 10%; the gas yield was 37–55% across the four tested feedstocks, while char covered 35–45% in the experiments performed using subcritical water without a catalyst by Wang and colleagues [104].

When rice straw was liquefied, however, a greater bio-crude yield of 21.1 wt% was attained in ethanol at 350 °C [66]. Gholizadeh and colleagues understood that biochar would support the hydrotreatment process through its unique structural properties [20]. To produce bio-oils, Wang and colleagues investigated the effects of solvents (water, ethanol, acetone, and carbon dioxide) on the liquefaction of pinewood sawdust. The experiment outcomes demonstrated that, by increasing the liquid yield and reducing the production of solid residue, both the catalyst and the solvent could significantly enhance the liquefaction process. The solvent had a significant impact on how the liquid products were distributed as well [105]. The direct liquefaction of woody feedstock using Ba(OH)_2_ as a catalyst greatly increased the yield of heavy oils by 50% [105,106]; Lu and colleagues examined the effects of cellulose HTC with varying initial concentrations of basic and acidic conditions using H_2_SO_4_, HCl, Ca(OH)_2_, and NaOH. Additives sped up the conversion of glucose as the concentrations increased and sped up the dissolution of the solid cellulose [107]. Additionally, acids promoted dehydration, which continued as the main carbonization mechanism with a lower oxygen content. Acid additives enhancing the production of CO_2_ promoted the breakdown of organic acids through decarboxylation [107].

To further elaborate the above discourse, Table 3 provides a comprehensive characterization of hydrochar properties produced under different severity conditions of HTC, along with the use of various catalysts and feedstocks. Feedstock collection of interest included the likes of *Spirulina platensis*, *Nannochloropsis*, straw, *Dunaliella tertiolecta*, sludge (sewage), food waste, pig feces, *Nannochloropsis* sp., Spruce Lignin, Spirulina, *Ulva prolifera*, bagasse, and Sunflower oil [1,23,30,46,69,71,108]. The higher heating values (HHV), measured in MJ kg^−1^, exhibit considerable variability, spanning from 15.5 to 39.6 MJ kg^−1^. The elemental analyses (wt%) of C, H, N, S, and O content showcase significant differences based on the catalyst, feedstock, and temperature. Various catalysts, including NiO, Ca_3_(PO_4_)_2_, Na_2_CO_3_, Fe, Mn, K_2_CO_3_, HCl, HNO_3_, H_2_SO_4_, Pt/C, ZSM-5, CH_3_COOH, MgMnO_2_, and HCOOH, contribute distinct impacts to hydrochar properties, influencing the elemental composition and HHV. Examples such as NiO with *Spirulina platensis* at 350 °C, yielding an HHV of 38.4 MJ kg^−1^, and Pt/C with *Nannochloropsis* sp. at 350 °C, producing a high HHV of 39.6 MJ kg^−1^, showcase the diverse outcomes achievable.

Wang and colleagues showed Ni/TiO_2_ as a better catalyst, enhancing the yield, quality, and carbonization conversion of biocrude. Ni/TiO_2_ was characterized by XRF, XPS, and XRD. The reaction temperature affected the HTL of microalgae Nano-chlorosis over Ni/TiO_2_, as 300 °C produced the highest liquefaction conversion of 89.28% with a maximum biocrude yield of 48.23% [71]. More so, the metals Fe, Ni, and Zn added to biomass via HTC were investigated elsewhere. Fe demonstrated the greatest performance with increased bio-crude production from 17.4% of the blank test to 26.5%, and an increase in the higher heating value (HHV) from 27.0 MJ/kg to 29.7 MJ/kg. Also, Zn increased the number of water-soluble products by slightly increasing the amount of bio-crude. The H/C ratio and HHV of the resulting biocrude were dramatically increased with each of the evaluated transition metals [72]. Abdullah and colleagues used HTC to produce an activated carbon catalyst from renewable mesocarp fiber obtained from palm oil. In their investigation, they found that adding K_2_CO_3_ and Cu(NO_3_)_2_ created a bifunctional catalyst that could be used to convert spent cooking oil into biodiesel. The catalyst had a mesoporous structure with a BET surface area of 3909.33 m^2^/g and an ideal treatment ratio of 4:1 (K_2_CO_3_:Cu(NO_3_)_2_). This resulted in elevated basic (5.52 mmol/g) and acidic (1.68 mmol/g) concentrations on the catalytic surface, which encouraged transesterification and esterification reactions [18]. Indeed, Table 4 provides a comprehensive overview of how different catalysts, including K_2_CO_3_, KOH, CaO, MnO, Na_2_CO_3_, TiO_2_, ZrO_2_, Na_2_CO_3_, Ni, among others [23,29,30,31,73,98,112,113], impact the HTC process, influencing product properties such as oil yield, gas yield, and chemical composition. Watanabe and colleagues studied the effects of the homogeneous and heterogeneous catalysts (H_2_SO_4_, NaOH and TiO_2_, ZrO_2_, respectively) on glucose in hot compressed water at 200 °C using a batch-type reactor. In their findings, the homogeneous catalyst demonstrated that the acid catalyst promoted dehydration, while the isomerization of glucose to fructose was catalyzed by alkali. Additionally, it was discovered that ZrO_2_ functioned as a base catalyst to enhance the isomerization of glucose, whereas TiO_2_ acted as an acid catalyst to promote the formation of 5-hydroxymethylfuraldehyde [27]; the application of K_2_CO_3_ on the HTC of wood biomass at 280 °C for 15 min decreased the hydrochar yield and the obtained oil contained mainly phenolic compounds [31]; Song and colleagues reported that bio-oil increased to 47.2% with 1.0 wt% of Na_2_CO_3_ from 33.4% without a catalyst in the conversion of corn stalk at 277–377 °C [29]. The importance of using catalysts in HTC is emphasized by their varied effects on different feedstocks and temperatures. For instance, K_2_CO_3_ and KOH at 550–600 °C favor a water–gas shift. Yim and colleagues investigated the effect of metal oxide catalysts like CaO, MgO, MnO, SnO, ZnO, CeO, NiO, AlO, and LaO on the supercritical HTC of empty fruit bunch (EFB) obtained from oil palm residues for the bio-oil yields and characteristics studied. EFB, water, and 1.0 wt% metal oxide were placed into a batch reactor and heated to 390 °C at a reaction time of 60 min. In their study, among the tested catalysts, the four most active metal oxides with lower electronegativity (CaO, MnO, La O, and CeO) provided a maximum relative yield of bio-oil, at 1.40 times that without catalyst [73]. The enumeration of a wide range of catalysts underscores their significance in tailoring the HTC process for sustainable biofuel and chemical production.

## 5. Catalyst Effect on the Morphological and Textural Character of Hydrochar

Investigating the changes in the specific surface area of HTC products, in particular hydrochar, could be achieved using the multi-point BET adsorption method [122]), while the structure, composition, and texture of the prepared catalysts and produced hydrochar were examined using scanning electron microscope (SEM). Numerous studies have investigated the impact of the addition of catalysts on the hydrochar properties of specific feedstock. According to Zhao and colleagues, the total surface area and pore volume of pure pomelo peel biochar were 6.7 m^2^/g and 24.4 mm^3^/g, respectively. With the addition of KOH as an activating agent, the surface area significantly increased (from 6.7 to 278.2 m^2^/g), as well as the pore volume (from 24.4 to 154.2 mm^3^/g) [91]. Elsewhere, four different kinds of activated carbons were employed, namely FeCl_2_, FeCl_3_, FeC_2_O_4_, and FeC_6_H_5_O_7_. Thus, in the context of the external surface area and total pore volume, FeC_6_H_5_O_7_-prepared activated carbon showed the highest results, while it was reported that FeCl_3_ and FeCl_2_ make promising substitutes for the production of high-quality activated carbon with a relatively high specific surface area and advantageous surface functional groups. The total surface area and micropore volume values were higher than those of the activated carbons obtained with FeC_2_O_4_ and FeC_6_H_5_O_7_ [123].

SEM is typically used to examine the morphology of hydrochars under various HTC conditions, producing micrographs that display the material’s physical characteristics and surface morphology [52]. Abdullah and colleagues’ study used mesocarp fiber (MF), employing HTC for pretreatment in the presence of H_3_PO_4_. Thus, activated carbon undergoes a modification process with Cu(NO_3_)_2_ and K_2_CO_3_ after carbonization. Following that, they utilize SEM images captured at 20,000× to identify changes in the hydrochar surface morphology. Consequently, their results of raw MF and impregnated MF hydrochar indicated a dense surface with few variable pore sizes, and significant changes were observed. They concluded that the pretreatment with HTC in the presence of H_3_PO_4_ improved the degradation of MF by cleaving long-chain compounds [18]. Additionally, another finding reported that the number of surface pores gradually increased in the HTC conditions, especially the temperature, and a mesh structure appeared before transforming into a bar-like structure [124]. Furthermore, SEM images of biochar were also investigated elsewhere, where the morphology was heterogeneous, with particles ranging from a few micrometers to agglomerates higher than 100 µm. A single particle detail was noticed, revealing an increased porosity. The larger surface porosities were not very deep; however, they were formed by several small pores, according to their findings. The BET results are consistent with the conclusion that these micropores could help in the adsorbing properties of this material [125]. For example, Table 5 showcases the BET analyses of diverse feedstocks, such as corn straw, mesocarp fiber, sludge, tobacco stems, cattail leaves, arundo donax linn, wheat straw, cornstalk, manure, rice husk, pomelo peel, and bagasse of sugarcane [91,108,126,127,128,129,130,131]. This table details the HTC process, presenting crucial information, including temperature, BET surface area, SEM, pore volume, and citations for each catalyst–feedstock pairing. These data provide insights into the specific surface characteristics of feedstocks under various catalysts and temperatures, which is crucial for understanding and optimizing the HTC process. Recently, increasing evidence obtained via characterization techniques reveals that the structural evolution of catalysts caused by the interplay with electrolytes, electric fields, and reactants brings about the formation of real active sites. Therefore, key ideas related to structural evolution, such as stability, active sites, catalysts, and their significance, are presented in this review. Furthermore, previous studies indicated that the presence of hydrophilic functional groups on the surface of the biochar may aid in the adsorption of hydrophilic reactants like ethanol. Reactants can easily reach the active sites due to a large pore size. In some circumstances, the advantages of biochar’s superior pore morphology may outweigh the drawbacks of its low –SO_3_H group density [40,88].

The hydrochar produced under various process conditions and from various feedstock types has a different structural characteristic. The majority of studies use FTIR techniques to examine the surface functional groups of raw biomass and hydrochar due to the complex composition of the hydrochar and the highly variable hydrothermal parameters and feedstock used in HTC. The Fourier infrared spectroscopy (FTIR) measurements were conducted to observe the changes in the functional groups of hydrochars’ properties following HTC under different conditions [122]. The functional groups available on the hydrochar and feedstock are usually determined using FTIR, as presented in Table 6, including an asymmetry stretching vibration, aliphatic hydrocarbon chain stretching, and bending vibrations [18]. Indicators and their corresponding functional groups and vibrations can be seen in Table 6. The symmetrical and asymmetrical C–H stretching vibrations of the methyl and methylene groups are significant. The absorption at 1717 cm^−1^ is attributed to C-O stretching, which may originate from carboxylic, ketones, or aldehydes acids. However, the carbonyl in carboxylic acids absorbs much more intensely than those in ketones and aldehydes, which, combined with the presence of –OH, can confirm the existence of carboxylic acids rather than ketones or aldehydes. The vibrations of the aromatic ring breathing cause absorption at 1612 cm^−1^ [133].

The functional groups associated with catalysts, their corresponding feedstock, and working parameters are shown in Table 6. Across the functional groups, there are single- and double-bond organic structures/stretched vibrations [38,41,69,91,102,131,134]. According to Sliz and Wilk’s findings, bands in the 800–900 cm^−1^ range represent C–H in the plane bend. The comparison of the methyl band at 1380 cm^−1^ and the methylene band at about 1470 cm^−1^ indicates that, after Virginia mallow undergoes HTC treatment, the branched-chain tends to become a more linear structure [122]. Similar findings reported that the bagasse from sugarcane demonstrates a characteristic absorption of herbaceous biomass. The band at 3419 cm^−1^ represented the characteristic absorption of -OH stretching vibration. The peaking at 2922 cm^−1^ was assigned to the symmetric methyl group [67]. Absorption at 1464 cm^−1^ and 1424 cm^−1^ indicate the C–H stretching of alkanes. Primary, secondary, and tertiary alcohols’ C–O stretching is the cause of absorption at 1121 cm^−1^ and 1099 cm^−1^. However, in the case of bio-crudes derived from de-ashed barks, absorption at 1717 cm^−1^ is significantly weaker, suggesting that the bio-crude contains either no carboxylic acids or very little due to the direct liquefaction of the barks. Since alkali compounds (K_2_CO_3_ or Ca(OH)_2_) are thought to catalyze the formation of carboxylic acids in the hydrothermal liquefaction of biomass, this result could be explained via the absence of alkali compounds in the bark ash [101], The bond vibrations corresponding to the C-H bending (1098 cm^−1^), CH_2_ rocking (719 cm^−1^), and C=O stretching (1737 cm^−1^) of hydrocarbons, free fatty acids, and esters are crucial [126]. Recently, research by Aysu and Halil demonstrated that the C-O stretching vibration bands at 1023.93 cm^−1^ and O-H stretching vibration bands at 3344.57 cm^−1^ disappeared in bio-chars [41], which could be attributed to the decomposition of raw feedstock and the removal of oxygen due to the thermal cracking of feedstock components producing carbonaceous hydrochar during the HTC process [108].

## 6. Areas of Future Research

Researchers who are new to thermal conversion technologies may consider this review as a foundation to rapidly deepen their understanding of catalyst applications/research. Moreover, areas of future research involving the application of catalysts in HTC treatment can be seen in Figure 3, which involve considerations like environmental, economical, analytical methods, feedstock/catalyst selection, as well as (HTC) operating conditions. Although several catalyst applications in thermal conversion technologies and approaches have been investigated in recent decades, the economic part of thermal technology does not seem thoroughly explored. Besides the economic aspect, there is a need for a broader scope of hydrochar-produced entities alongside catalyst functioning via HTC, especially for activated carbon production. Such studies should employ optimization operating/process conditions, which should allow for a combination of feedstock, HTC catalysts, and activation procedures—all of which are aimed at enhanced supercapacitor performance. In addition to the performance, the following environmental and recycling perspectives could be considered: (a) environmental impact and cost, which might necessitate the testing of novel activating agents and HTC catalysts with less environmental impact, and (b) the recycling of the activation agent and HTC liquid fraction where possible [35,135]. Feasibly, the massive body of scientific evidence regarding hydrochar reveals that the liquid/gas product needs additional attention. More detailed analyses of by-products that contain intermediate products would elevate the understanding of hydrothermal conversion, especially the formation of hydrochar. Moreso, the prior-treatment process is essential to the HTC liquid fraction and hydrochar for nutrient substances, such as P and N recovery. For emphasis, the distribution, transformation mechanism, recovery, and future initiatives should also seriously consider the relevant analysis and treatment [52,67]. From an environmental perspective, HTC products offer numerous benefits for their applications. However, the economic aspects need to be estimated based on the investments in the conversion technology and production costs [35]. This could be achieved through the addition of a suitable catalyst to a specific feedstock regarding the target product application.

Again, as shown in Figure 3, areas of future research involving the application of catalysts in HTC treatment have a direct/indirect link to life cycle assessments (LCAs), which can assist in evaluating and comparing various scenarios of renewable process integrations, providing answers to concerns of society and decision makers. Biochar is used as a heterogeneous catalyst or support because it is inexpensive, has a large surface area, can be tailored to specific functional groups, and is perfect for producing biodiesel. Because of its stable structure, strong mechanical and thermal stability, and chemically hierarchical structure derived from biomass, biochar is regarded as a superior catalyst in a variety of catalytic applications [35]. When catalysts are applied in HTC, the water biomass mixture’s properties are first affected, leading to the intended modifications in the process and end products. Consequently, the catalyst of choice is determined by the user’s ultimate goal. For instance, the use of acid catalysts that promote hydrolysis can enhance the production of hydrochar. On the other hand, by using basic catalysts to promote the formation of liquid products, the formation of hydrochar can be reduced [136]. Furthermore, catalysts can also be used to reduce emissions like NOx that are produced during biomass combustion. In addition to the high-temperature thermal fixation of nitrogen in the combustion air, which contains excess oxygen, NOx is also produced by the fuel’s chemically bound nitrogen being converted. Thus, the fuel’s N content is one of the variables that influences the quantity of NOx generated. The behavior of N during the HTC treatment is related to the precombustion control of NOx, and it has been demonstrated that utilizing catalysts to switch from pure water to a more basic and acidic aqueous solution can enhance the removal of N [137]. Mumme and colleagues investigated the effects of cellulose and an agricultural digestate on natural zeolite in HTC. Zeolite significantly and marginally increased the energy and carbon content of the hydrochar that was produced from digestate and cellulose. Moreover, the catalytic HTC products had larger pore volumes and surface areas. The primary cause of the variations observed between the digestate and zeolite results is the zeolite layer’s physical and chemical shielding of organic compounds such as cellulose. The fact that zeolite retained the digestate’s cellulose fraction resulted in this [138]. According to research from Abd Hamid and colleagues, complete carbonization in HTC can occur at temperatures as low as 200 °C when Lewis acid catalysts (FeCl_2_ and FeCl_3_) are used. Prior research has examined the impact of catalysts, including acetic acid, KOH, KCl, Na_2_CO_3_, and NH_4_Cl, on the hydrochar of HTC [139,140].

However, careful catalyst development is required to scale up the catalytic process. If premature deactivation is to be prevented, then a better understanding of the process design must be applied. Ahamed and colleagues carried out a comparable LCA through contrasting an incineration system and an AD system with a combined HTC system and oil refinery system (for the transesterification of the HTC liquid fraction with acid treatment, which produces glycerol and bio-diesel). In their study, they took into account one ton of food waste, the system boundary, which included collection, processing, waste conversion, and the disposal of food waste, as well as three outputs, namely electrical energy (using biodiesel with a 35% efficiency), hydrochar, and glycerol. According to their LCA results, when the feedstock’s oil content exceeds 5%, the suggested HTC and refinery combination is more advantageous [53]. This review found that the addition of a catalyst in the hydrothermal process to enhance the process performs better technically and has a smaller environmental impact. Overall, it can be said that defining the boundaries of the scenarios while taking into account the difficulties is critical to take into account in a thorough LCA of the use of catalysts in HTC. This perimeter encompasses not only the stages of preparation and transformation, but also the post-treatment problems brought on by HTC, like waste management and expensive energy requirements.

## 7. Concluding Remarks

Critically, there are competing variables of influence involving HTC products, like gas formation, hydrochar, and liquid fraction. Also, the physicochemical components of the analysis can include ultimate or proximate analysis, aromatic structure, surface functional groups, and morphological aspects emanating from catalyst additives. Understanding how biomass behaves under hydrothermal conditions helped by various alkaline/transition metals is an essential first step to the complicated liquefaction mechanism of lignocellulosic biomass, as well as how it affects the products’ quality and yield. Indeed, multiple catalysts and thermal operating conditions would allow for the production of lignocellulosic HTC products, most of which have been primarily empirical with an emphasis on rapid commercial process development. Fundamentally, biomass conversion should project a high-quality yield and by-products including proteins, lipids, residual carbohydrates, fibers, fat, and other biopolymers that are efficiently transformed into biofuels and biochemicals through waste-to-energy technologies.

For emphasis, the application of catalysts should enhance the degree of carbonization and surface modification, alongside the introduction of better heteroatoms, which should substantially improve the effectiveness of activated carbon. Overall, the addition of catalysts to the HTC, with a secondary treatment stage if necessary, would potentially resolve the existing barriers and produce activated carbon with special qualities that go above and beyond the current standards. To establish the use of homogeneous catalysts in HTC would be crucial in promoting the success of bio-oil production, which can be attributed to the decrease in tar and char formation, with K_2_CO_3_ and NaOH being the most and least effective catalysts, respectively. On the contrary, heterogeneous catalysts have unstable effects, while alkali and alkaline earth metals tend to increase the reaction rate in gasification. Nickel is the most effective catalyst for tar reduction in the gasification process. The direction of future work should look at the challenges associated with low-cost catalysts, as well as those of the best quality, being employed in thermochemical conversion technologies.

## Figures and Tables

**Figure 1 materials-17-02579-f001:**
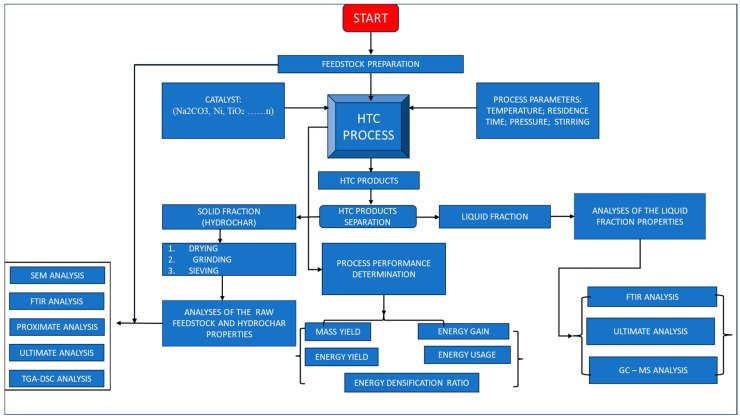
Key stages in the application of catalysts in the thermal conversion process, from feedstock selection, catalysts types, treatment methods, and analytical methods of output.

**Figure 2 materials-17-02579-f002:**
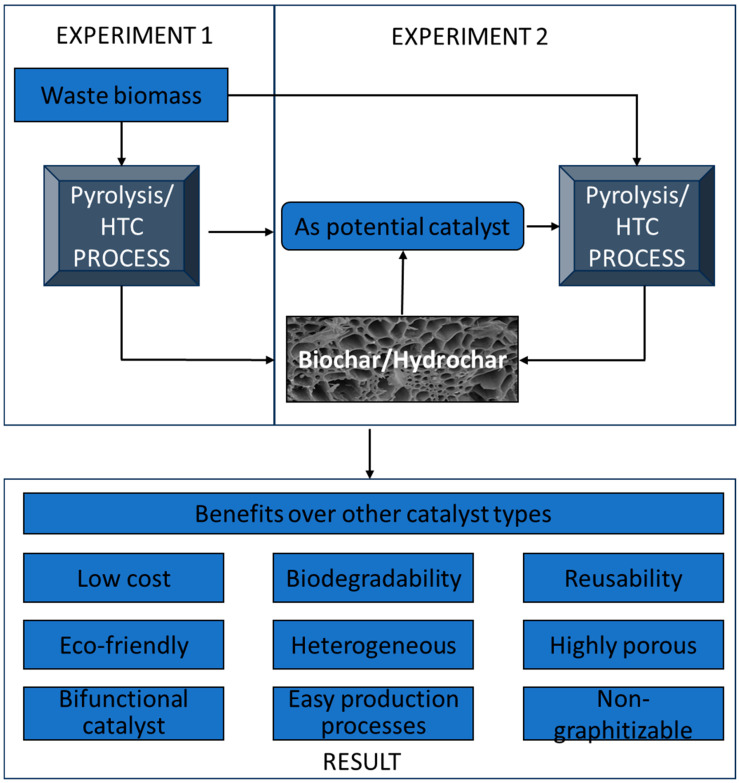
The benefit of the application of biochar as a potential catalyst.

**Figure 3 materials-17-02579-f003:**
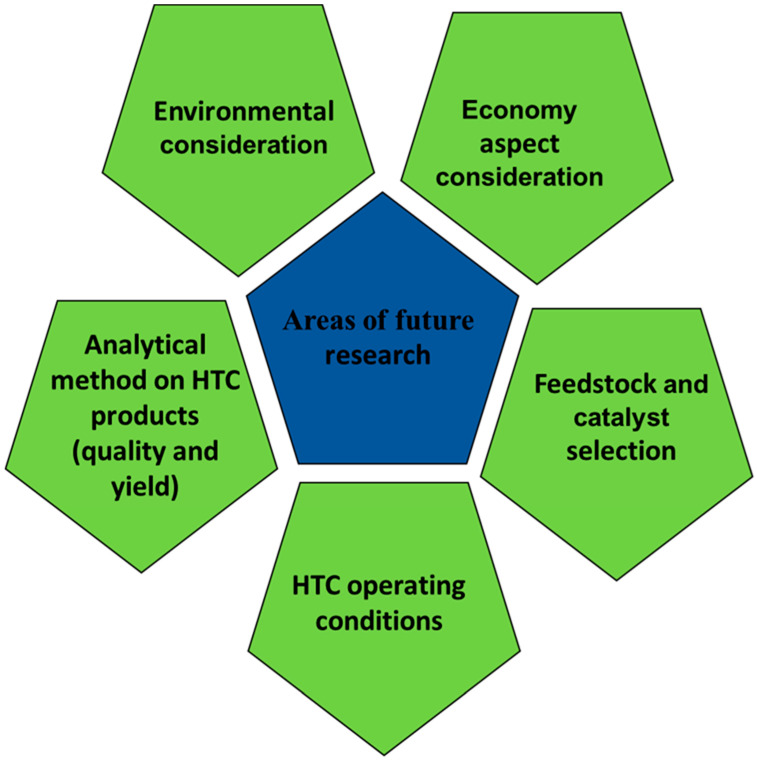
Areas of future research involving the application of catalysts in HTC treatment.

**Table 1 materials-17-02579-t001:** Summary of reviews involving hydrothermal treatment, biomass conversion, and catalysts/catalytic processes from the past decade.

Review Objective	Key Sections	References
Reviewed biochar value of catalysts in biofuel production, alongside the processes utilized/various biomass sources	- Methods of biochar production - Biochar composition - Biochar-based catalysts - Biochar as a catalyst for fuel production	[43]
Review summarized/critically discussed catalyst types/catalytic mechanisms, as well as process parameters	- Impact of process parameters on hydrothermal carbonization - Reaction pathways that the feedstock compounds take during hydrothermal carbonization - Progress of catalytic hydrothermal carbonization - Hydrochar application/catalyst selection - Environmental/techno-economic features of catalytic hydrothermal carbonization - Challenges/future prospects of catalytic hydrothermal carbonization	[46]
Reviewed biochar as a catalyst for biomass conversion via thermolysis (pyrolysis)/hydrothermolysis (liquefaction/gasification).	- Biochar - Use of biochar in catalysis - Biochar-based catalytic biomass conversion processes	[44]
Reviewed whether biochar and hydrochar are sustainable catalysts for persulfate(PS) activation	- PS activation mechanism - Properties desired in hydrochar/biochar for PS activation - Strategies for desired char properties - Whether biochar/hydrochar is a sustainable catalyst for persulfate activation	[45]
Reviewed catalysts for high bio-oil yields with improved quality/factors that influence the catalytic hydrothermal liquefaction (HTL), mechanisms of catalytic-HTL reaction, HTL products	- Catalytic effect on bio-oil yield - Use of catalysts in HTL - Mechanism - Physicochemical properties of catalytic bio-oil - Effect of catalysts on the aqueous phase extract of HTL - Effect of catalyst on the gas fraction of HTL - Effect of catalyst on HTL biochar	[47]
Reviewed hydrochar characteristics/reaction mechanisms for char production technology, e.g., hydrothermal carbonization, hydrochar activation and functionalization	- Hydrochar versus biochar - Hydrochar production technologies - Activation and functionalization of hydrochar - Applications of activated hydrochar	[37]
Reviewed conversion techniques that transform lignocellulosic biomass waste into biochar (gasification and pyrolysis), compared conversion techniques in terms of benefits, drawbacks, and limitations	- Biomass conversion techniques - Biochar modification techniques - Applications of biochar-derived catalysts - Impacts of industrial revolution 4.0 on biomass industry	[49]
Reviewed biochar-based catalysts for fuel production, thermochemical routes and their yield, composition/production, and choice for fuel production	- Techniques for biochar production - Composition of biochar - Why biochar-based catalysts? - Biochar catalysts utilized to produce fuel	[35]
Reviewed two strategies to convert biomass into functional catalysts (Photocatalytic/Nonirradiant application of biomass)	- Biomass conversion to hydrothermal carbonation carbon (HTCC) catalysts - Biomass conversion to a biochar catalyst - Differences between biochar and HTCC	[50]
Reviewed catalysts’ effects on thermochemical conversion research/development involving biomass/thermochemical conversion processes	- Torrefaction - Pyrolysis - Liquefaction - Gasification	[28]
Review summarized preparation/modification/catalytic application of biochar in biofuel production, from biomass hydrolysis to tar reduction	- Biochar synthesis - Biochar characteristics - Biochar modifications - Biochar-based catalysts to produce biofuel	[40]
Reviewed HTL catalytic upgrade/catalytic performances on algae (HTL/biocrude) upgrade	- Catalytic HTL involving algae - Catalytic upgrade involving biocrude - Reaction mechanism in algae HTL/biocrude upgrading	[48]
Reviewed the research progress of heterogeneous catalysts for biodiesel production/low grade feedstocks	- Problems of currently used catalysts for biodiesel - Advantages of solid acid catalysts - Effects of some reaction parameters for biodiesel production	[19]
Reviewed versatile applications of biochars as catalysts that upgrade biomass	- Thermochemical degradation to form biomass/biochar - Activation/functionality of biochars as catalysts/catalyst support - Biochar-based catalysts that upgrade biomass	[51]

**Table 2 materials-17-02579-t002:** Catalyst (homogeneous and heterogeneous) types by feedstock, HTC operating conditions, and product yield.

Catalyst	Type	Feedstock	HTC Operating Conditions	Products Yield (%)			Reference
				Bio-Oil	Char	Gas	
**Heterogeneous Catalyst Systems**							
H-ZSM−5	Acidic catalyst	Algae	70 mL water, 7 g algae, catalyst of 0.35 g, at 300 °C for 20 min	34	24	42	[80]
H-ZSM−5		Wheat straw	350 °C for 60 min, catalyst to biomass—0.1:1	28	37	35	[28]
H-ZSM−5		Wheat husk	350 °C, catalyst to biomass—0.1:1, 1 h	26	31	43	[74]
Ce/H-ZSM−5		Algae	70 mL water, 7 g algae, catalyst of 0.35 g, at 300 °C for 20 min	50	18	32	[80]
CaO	Basic catalyst	Fruit	390 °C for 30 min, water used was at a ratio of 1:10 of biomass and 1 wt% catalyst	63	-	-	[73]
		bunch					
Pd/C	Metallic catalyst	Algae	87.5% water volume, at 350 °C for 60 min, 15 mg of catalyst	38	-	-	[81]
CoMo/Al_2_O_3_		Algae	95% water volume, at 350 °C for 60 min, 0.38 g of catalyst	55	-	-	[68]
Ni/SiO_2_-Al_2_O_3_		Algae	95% water volume, at 350 °C for 60 min, 0.384 g of catalyst	55	-	-	[68]
Ni/TiO_2_		Algae	480 g of water, at 300 °C for 30 min, and a catalyst of 10% of algae of 120 g	31	-	-	[71]
Pt/C		Algae	350 °C for 60 min, 95% water volume, and 0.38 g of catalyst	49	-	-	[68]
Ni		Cellulose	300 °C for 10 min, cellulose (1 g), water (5 g) and Ni (0.1 g)	25	6	13	[72]
Zeolite	Neutral catalyst	Algae	95% water volume, at 350 °C for 60 min, 0.38 g of catalyst	48	-	-	[68]
MgMnO_2_		Bagasse	250 °C in 1 to 15 min, catalyst 2 g, and 20 g of biomass	60	12	28	[67]
**Homogeneous Catalytic Systems**							
KOH	Basic catalyst	Algae	350 °C, 3 g algae with 27 mL of catalyst	15	5	10	[65]
Na_2_CO_3_		Algae	300 °C for 30 min, 20 g algae with 150 mL water, 5 wt% catalyst	21	20	30	[75]
Na_2_CO_3_		Algae	250 °C for 60 min, 10 g of algae with 1:6 of biomass-to-water	38	25	8	[25]
K_2_CO_3_		Sewage sludge	350 °C, 7 g of sludge, 2% weight of sludge	45	7	-	[77]
CH_3_COOH	Acidic catalyst	Algae	350 °C, 3 g algae, and 27 mL of catalyst	17	5	25	[65]
H_2_SO_4_		Algae	290 °C for 20 min, algae 30 g with 1:3 of biomass-to-water	28	12	60	[76]
HNO_3_		Food waste mixture	250 °C for 120 min, feedstock 35 g, 350 mL water, catalyst 10% of biomass	-	47	-	[79]
FeSO_4_		Sewage sludge	300 °C for 40 min sludge to water 1:5, catalyst, and 5 wt.% of dry Sludge	48	-	-	[78]
FeSO_4_		Pine wood	350 °C for 40 min, 1 g of wood, 2% weight of wood	63	-	10	[82]

**Table 3 materials-17-02579-t003:** Characterization of hydrochar properties produced in the severity of HTC conditions.

Catalysts	Feedstock	Temp. (°C)	HHV (MJ kg^−1^)	Elemental Analysis (wt%)					Reference
				C	H	N	S	O	
NiO	*Spirulina platensis*	350	38.4	75	9	6	1.4	6.5	[23]
Ca_3_(PO_4_)_2_			35.1	72	9	4	1.1	12.7	
Na_2_CO_3_			36.3	72	9	5	0.9	11.8	
Fe	Nannochloropsis	300	35.5	70	9.8	7	0.4	12.2	[71]
Mn			33.2	69	8.6	7.2	0.4	14.6	
K_2_CO_3_	Straw	300	17.2	53	4.3	0.9	0.7	40	[30]
Na_2_CO_3_	*Dunaliella tertiolecta*	360	30.7	63	7.7	3.7	-	25.1	[69]
HCl	Sludge	230	-	46	4.8	3.7	0.1	19.2	[46]
HNO_3_	Food waste	250	-	57	5.8	1.6	0.5	23.4	
H_2_SO_4_	Pig feces	230	-	56	4.2	2.4	-	36.8	[1]
Pt/C	*Nannochloropsis* sp.	350	39.6	75.9	10.8	4.0	0.7	8.48	[71]
K_2_CO_3_	Straw	300	27.2	67.9	7.6	0.8	0.6	23.2	[30]
ZSM-5	Spruce lignin	-	-	64.7	6.3	0.5	0	28.6	[109]
CH_3_COOH	*Spirulina*	-	35.1	71.7	9.7	6.1	0.9	11.6	[110]
H_2_SO_4_	*Ulva prolifera*	180	15.5	35.7	6.5	2	2.2	32.4	[111]
MgMnO_2_	Bagasse	250	32.6	65.9	10.2	0.4	0.3	23.3	[67]
K_2_CO_3_	Sewage sludge	350	36.6	75.6	10.6	4.7	-	9.2	[77]
HCOOH	Sunflower oil	350	37.3	68.4	11	0.2	0	20.5	[22]

**Table 4 materials-17-02579-t004:** List of catalyst types and effects in the HTC process on product properties.

Catalysts	Feedstock	Temperature °C	Effect	Reference
K_2_CO_3_ and KOH	Organic wastes and wet biomass	550–600	Water–gas shift	[114]
MnO, CaO, CeO_2_, MgO, SnO, Al_2_O_3_, NiO, La_2_O_3_	Empty fruit bunch	390	Addition of CaO, CeO_2_, MnO, and La_2_O_3_ catalysts maximized bio-oil yield	[73]
Na_2_CO_3_	Cornstalk	277–377	Oil yield increased	[29]
K_2_CO_3_	Wood biomass	280	Decreased the char yield	[31]
K_2_CO_3_	Barley straw	280–400	Oil yield increased	[30]
Ni, Na_2_CO_3_	Cellulose	200–350	Char decreased	[115]
Ni, K_2_CO_3_	Glucose	350–500	Water–gas shift	[113]
H_2_SO_4,_ NaOH, ZrO_2,_ TiO_2_	Glucose	200	Isomerization of glucose increased	[27]
Na_2_CO_3_ NiO	*Spirulina platensis microalgae*	300–350	Increased oil yield	[23]
NiO, Ca_3_(PO_4_)_2_	*Spirulina platensis microalgae*	300–350	Increased gas yields	[23]
Ni/TiO_2_	*Nannochloropsis microalgae*	300	Increased hydrocarbons in bio-oil and acids	[71]
Pd/HZSM-5@meso-SiO_2_	*Spirulina microalgae*	380	Oil yields increased and reduced coke yields	[116]
Co-Zn/HZSM-5	Pine sawdust	300	Hydrocarbon content and oil yields increased	[114]
Na_2_CO_3_	*Pavlova microalgae*	250–350	HHV and oil yields increased	[25]
MgMnO_2_	Sugarcane bagasse	250	Degradation of lignin	[67]
Ni	Cellulose	350	Enhanced H_2_ yield	[117]
H_2_SO_4_, zeolite, FeS	Wheat straw	100–180	Degradation of lignin	[118]
H_2_SO_4_	*Ulva prolifera*	180	Increased oil yields	[111]
K_2_CO_3_	Sewage sludge	350	Promote the hydrolysis of carbohydrate to increase the oil yield	[2]
Biochar at 875 °C + KOH	Woody biomass	-	Surface area of hydrochar increased	[35]
Biochar at 875 °C + KOH	Rice husk	-	Surface area of hydrochar increased	[119]
Biochar at 875 °C + KOH	Pomelo	-	Increase the surface area of hydrochar	[91]
Biochar + sulfonated with SO_3_H	Wood	-	The porosity and surface area of the biochar increased	[96]
Ru/C	Oil from beech wood	350	High HHV of oil and low oxygen content	[120]
Ru/TiO_2_	Oil from beech wood	350	It improves the oil yield	[120]
Pd/C	Oil from beech wood	250	Demonstrate a high oil yield and reduced oxygen content	[120]
Pt/C	Oil from beech wood	250	High oil yield, but oxygen content is relatively high	[120]
Fe	Cellulose	300	(HHV) increased from 27.0 to 29.7 MJ/kg of the blank test and the bio-oil yield from 17.4% to 26.5%	[72]
Zn	Cellulose	300	A slight increase in the bio-oil yield and water-soluble products also increased	[72]
Fe	Biomass	340	Less gas emission for obtained HTL bio-jet fuel and lower production costs	[121]

**Table 5 materials-17-02579-t005:** BET analyses of selected feedstock and different catalyst types.

Catalyst	Feedstock	Temp. (°C)	Surface Area (m^2^/g)	SEM (nm)	Pore Volume (cm^3/^g)	Reference
KOH	Pomelo peel	500	278.2	5000	154.2	[91]
ZnCl_2_	Corn straw	200	110.2	10,000	0.6867	[126]
2K_2_CO_3_/CuO	Mesocarp fiber	200	678.8	5000	0.494	[18]
TiO_2_	Sludge	-	-	500	-	[127]
K_2_CO_3_	Tobacco stems	450	255.7	-	1.647	[128]
H_2_SO_4_	Cattail leaves	200	423.0	20,000	0.286	[129]
FeCl_3_	Arundo donax Linn	-	927.0	5000	0.509	[123]
FeCl_2_	Arundo donax Linn	-	760	5000	0.466	[123]
ZnCl_2_	Wheat straw	200	106.1	10	0.6195	[126]
SO_3_H	Cornstalk	400	20.58	-	0.03	[131]
HCl	Manure	190	28.92	-	0.088	[130]
Ru	Rice husk	520	806	-	0.58	[119]
Biochar + SO_4_	Wood	400	242	-	0.13	[95]
Citric acid	Pomelo peel	200	11.72	1000	0.06	[132]
Fe	Bagasse of sugarcane	200	75	-	-	[62]

**Table 6 materials-17-02579-t006:** Functional groups associated with catalysts, their corresponding feedstock, and working parameters.

Catalyst	Feedstock	Temp (°C)	Wavenumber (cm^−1^)	Functional Group	Reference
SO_3_H	Cornstalk	400	1177 and 1043	O=S=O asymmetric stretching	[131]
Graphene oxide	Tobacco		2800–3000	C–H aromatic structure and stretching vibration of aliphatic	[134]
ZnCl_2_	Sunflower	600	3700 and 3000	C-H aliphatic stretching vibration	[38]
K_2_CO_3_	Switchgrass	235	1166	C-O-C asymmetry stretching of hemicelluloses and cellulose	[103]
Na_2_CO_3_	Microalgae	360	1269 and 967	C-O Stretching	[69]
KOH	Palm fruit bunch	270	1680–1570	C-C stretching of aromatic groups	[102]
Ca(OH)_2_	Pine bark	300	1717	C=O stretching	[101]
Ni	Cellulose	300	3300	O–H stretching vibration of in phenols and alcohols	[72]
Fe	Paulownia wood	340	1700	Indicated the presence of ketone and C=O stretching vibration	[99]
K_2_CO_3_	Barley straw	300	1263, 1201, 1113 and 1032	The C-O stretching vibrations	[30]
Na_2_CO_3_	Spirulina	350	2935	Indicating C–H stretching vibrations bonds	[23]
8K_2_CO_3_/CuO	Mesocarp fiber		3102	V-OH stretching	[18]
Citric acid	Pomelo peel	220	2000–1000	Indicated the existence of C-C and C-O functional groups	[132]
H_2_SO_4_	Cattail leaves	200	2000–1000	OK group on the surface generated the -OH group	[129]
ZnCl_2_	Prosopis farcta	295	3344	O-H stretching vibration bands	[41]

## Data Availability

No data were used for the present study.

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
