# Peer review of "Catalyst-Enhancing Hydrothermal Carbonization of Biomass for Hydrochar and Liquid Fuel Production—A Review"

_materials, 2024, doi:10.3390/ma17112579_

Round 1
Reviewer 1 Report
Comments and Suggestions for Authors
The authors present their review work on “Catalysts impact on hydrothermal carbonization (HTC) for biomass conversion”, the topic is of interest. This review mostly focuses catalysts types, functionalization of biochar, process enhancement, hydrochar characteristics and its role as catalysts for thermochemical technologies. After careful consideration, I believed it is suitable for the publication in this journal. However, a few clarifications would enhance its impact.
Comments:
1. Could you provide figures for detailed explanation in the “Section 5”
2. Provide graphical abstract.
3. Avoid typological error such as spaces, subscripts, superscripts, etc.
4. Add biochar catalysts for biodiesel synthesis data in the Tables.
5. Include relevant recent references regarding biochar catalyst for biodiesel production in this field.
6. Follow the journal guidelines to prepare the manuscript.
7. Any potential limitations or challenges encountered in this process, and how were they mitigated?
8. What should the authors suggest to overcome the deactivation behaviors while designing the new catalyst?
9. Role of catalyst mechanism, active sites and stability should be discussed.
10. Moreover provide section regarding LCA.
Comments on the Quality of English LanguageThe author should be improved the language of the manuscript. Avoid grammatical mistakes.
Author Response
The answers to the reviewer's comments are in the attached file.

Reviewer 2 Report
Comments and Suggestions for Authors
The review undoubtedly deals with a topical and highly important issue in the context of so-called biorefinery related processes.
The review is comprehensive and covers various general aspects of the hydrothermal carbonisation process. The section on biochar production is very well summarised and developed. However, the section on catalytic systems could be improved so that the reader who intends to carry out the process in the presence of catalysts can design a material according to the desired output stream. Although there is valuable information in the review, it is difficult to extract on first reading. It is recommended that the discussion of point 2 be improved, sorted and expanded:
-Please sort Table 2 into homogeneous and heterogeneous catalytic systems.
-Please classify homogeneous catalytic systems into acidic and basic and identify catalysts with Lewis acid sites.
-Please classify catalysts as acidic, basic, metallic and/or bifunctional.
- Section 2 and Table 2. In thermochemical catalytic processes, the presence of catalysts, in addition to producing a different distribution of the yield towards each fraction (gas, liquid or solid), seeks to direct the transformation towards a particular family of compounds. It is proposed to add to Table 2 the main compounds found in the liquid and gas phases for each system. This point may be of interest to the reader, since it is not only the yield that is of interest, but also the selectivity of the catalytic systems.
In recent decades, ionic liquid-based catalysts have been used to assist the process. Could the authors add any comments on this?
-In the case of metal catalysts, please mention whether they affect the composition of the gaseous fraction. For example, is the H2/COX ratio in the exhaust stream improved?
The same comments can be used to explain the processes mentioned in Table 4. Please note that the analysis of the processes, efficiencies and their correlation with the catalytic sites is incomplete. Please correlate the performance of the catalytic processes used with the process temperature and the type of catalytic site (acidic, basic, metallic, etc.). What is the desired fraction in each reported work?
In line 188, please check the nomenclature of the salts.
Author Response

(The authors gave the same response as above.)

Reviewer 3 Report
Comments and Suggestions for Authors
Comments to the Authors
This paper discusses presents a review about hydrothermal carbonization of biomass for the production of liquid fuels and hydrochar. The authors focused on 4 main aspects: HTC catalyst types, biochar application, catalytic enhancement of HTC and hydrochar properties. This is an interesting document that is very well written, presented and organized. So, I recommend accepting this manuscript, leaving only some minor comments for the authors:
11. Revise the entire document in order to correct the chemical formulas. For example, both H2 in line 85 should be H2. Also, in line 85, TOC should also be subscript. Other examples are in lines 316, 319, 341, 343, etc.
22. In sentence 406-408, pomelo peel is mentioned twice.
3. In Table 5, the authors present a column with “SEM (nm)”, but lack to explain in meaning and which variable is being measured by SEM.
Author Response

(The authors gave the same response as above.)

Reviewer 4 Report
Comments and Suggestions for Authors
Ref.: Ms. No. materials-3003014
Title: Catalyst enhancing hydrothermal carbonization of biomass for hydrochar and liquid fuel production- A review.
Article Type: Review
Reviewer comment:
Reviewer: I appreciate the efforts of the author for the review article. This work is suitable for the materials journal. However, there are some comments to improve the quality of the article for the standard of the materials journal.
1) What are the biological catalysts used for transesterification reactions?
2) What are the functional groups present in biochar?
3) Please mention the diverse applications of biochar in the revised manuscript.
4) What is hydrothermal liquefaction process. Please describe it in the revised manuscript.
5) Ba(OH)2 should be Ba(OH)2.
6) How did you know that K2CO3 and Cu(NO3)2 exhibit mesoporous structure?
7) What is the effect of FeCl2 on activated carbon surface area? The author should discuss it in a revised manuscript.
8) Authors stated that FeC6H5O7 prepared activated carbon showed the higher external surface area. Later also stated FeCl3 modified activated carbon showed higher surface area than FeC2O4 and FeC6H5O7 modified activated carbon. Two statements are confusing. Please clarify it in the revised manuscript.
9) K2CO3 should be K2CO3.
10) Cm-1 should be cm-1 Please check the units of 1121, 1099 and 1717 bands in FTIR study
The quality of English is ok.
Author Response

(The authors gave the same response as above.)

Reviewer 5 Report
Comments and Suggestions for Authors
Please see the files.

Author Response

(The authors gave the same response as above.)

Round 2
Reviewer 2 Report
Comments and Suggestions for Authors
Most of my comments were accepted and comments were added to this new version. I suggest accepting the paper in this revised version